# The Risk Genes for Neuropsychiatric Disorders *negr1* and *opcml* Are Expressed throughout Zebrafish Brain Development

**DOI:** 10.3390/genes15030363

**Published:** 2024-03-14

**Authors:** Judith Habicher, Ilaria Sanvido, Anja Bühler, Samuele Sartori, Giovanni Piccoli, Matthias Carl

**Affiliations:** 1Department of Cellular, Computational and Integrative Biology, CIBIO, University of Trento, 38123 Trento, Italy; judith.habicher@unitn.it (J.H.); ilasanvido@gmail.com (I.S.); anja.buehler@uniklinik-ulm.de (A.B.); samuele.sartori@unitn.it (S.S.); giovanni.piccoli@unitn.it (G.P.); 2Molecular Cardiology, Department of Internal Medicine II, University of Ulm, 89081 Ulm, Germany

**Keywords:** *negr1*, *opcml*, IgLON, zebrafish development, brain, neuropsychiatric disorder

## Abstract

The immunoglobulin LAMP/OBCAM/NTM (IgLON) family of cell adhesion molecules comprises five members known for their involvement in establishing neural circuit connectivity, fine-tuning, and maintenance. Mutations in IgLON genes result in alterations in these processes and can lead to neuropsychiatric disorders. The two IgLON family members NEGR1 and OPCML share common links with several of them, such as schizophrenia, autism, and major depressive disorder. However, the onset and the underlying molecular mechanisms have remained largely unresolved, hampering progress in developing therapies. NEGR1 and OPCML are evolutionarily conserved in teleosts like the zebrafish (*Danio rerio*), which is excellently suited for disease modelling and large-scale screening for disease-ameliorating compounds. To explore the potential applicability of zebrafish for extending our knowledge on NEGR1- and OPCML-linked disorders and to develop new therapeutic strategies, we investigated the spatio-temporal expression of the two genes during early stages of development. *negr1* and *opcml* are expressed maternally and subsequently in partially distinct domains of conserved brain regions. Other areas of expression in zebrafish have not been reported in mammals to date. Our results indicate that NEGR1 and OPCML may play roles in neural circuit development and function at stages earlier than previously anticipated. A detailed functional analysis of the two genes based on our findings could contribute to understanding the mechanistic basis of related psychiatric disorders.

## 1. Introduction

The immunoglobulin LAMP/OBCAM/NTM (IgLON) protein family comprises the Opioid Binding Protein/cell Adhesion Molecule-Like (OPCML), previously named OBCAM or IgLON1 [1], Neurotrimin (NTM or IgLON2) [2], Limbic System-Associated Protein (LSAMP or IgLON3) [3], Neural Growth Regulator 1 (NEGR1, KILON or IgLON4) [4], and IgLON5 [5] (Table 1). All proteins are characterised by three Ig domains and a glycosylphosphatidylinositol (GPI) anchor [6]. IgLON neural adhesion protein family members form both homo- and heterodimers and have diverse roles in neural development, neurite outgrowth, neuronal arborisation, axon fasciculation, and synapse formation, including plasticity [4,7,8,9,10]. In addition, IgLON genes are tumour suppressors in a number of non-neural organs and tissue types [6,11,12,13,14,15].

Among the five IgLON family members, both NEGR1 and OPCML have been linked to major depressive disorder [16,17,18,19,20], schizophrenia [21,22,23,24,25,26], autism [27,28], anorexia nervosa [29,30], and Alzheimer’s disease [31,32]. Other pathologies and disorders ranging from dyslexia to Huntington’s disease and obesity were attributed to alterations in one gene or the other [33]. These findings suggest a role of NEGR1 and OPCML in partially overlapping brain areas. In vitro studies and loss-of-function investigations in mammalian animal systems have started to shed light on the mechanisms causing the human phenotypes [22,33,34,35]. However, much of the cellular and molecular basis underlying the genes’ role in these disorders in vivo has remained elusive. Moreover, available studies have focused on rather late stages of nervous system development and adulthood and the onset of the neuropsychiatric disorders linked to NEGR1 and OPCML has not been defined [4,10,36,37,38]. For instance, a detailed early gene expression analysis could reveal insights into the time window in which the genes begin to exert their function. Such studies can contribute to developing therapeutic options other than symptomatic treatments.

IgLON protein-encoding genes are evolutionarily conserved from arthropods to teleosts and mammals, including humans [39] (Table 1). The zebrafish (*Danio rerio*) genome contains all five family members, which, however, have not been studied extensively to date. The zebrafish is an excellent model for investigating gene functions in vivo and complements the methodologies used in other vertebrate model systems. Especially at early stages of development, the combination of transgenesis for the fluorescent visualisation of proteins and cells in normal and genetically manipulated transparent embryos and time-lapse analysis facilitates exploring gene functions under physiological conditions. Moreover, the abundance of ex utero developing small-size embryos allows for large-scale screening of compounds to elucidate the molecular underpinnings of diseases and to identify new therapeutic targets as well as ameliorating substances [40,41]. Drug discovery is further aided by the increasing number of robust behaviour test systems in particular for neuropsychiatric disorders [42].

Herein, we report the spatio-temporal expression of *negr1* and *opcml* during zebrafish embryonic development. We discovered that transcripts of both genes are maternally provided. Subsequently, *negr1* and *opcml* are expressed in brain regions similar to mammals, as well as in neural circuits not described in other vertebrates to date. Moreover, comparing *negr1* and *opcml* expression domains to each other at various stages of neural circuit formation reveals partial overlaps. Our analysis can serve as a starting point for functional in vivo studies to disentangle the involvement of IgLONs in common and distinct types of neuropsychiatric disorders.

## 2. Materials and Methods

### 2.1. Animals

Adult AB/TL wild-type zebrafish were kept under standard conditions of 13/11 h light/dark cycles at 28 °C [43]. Embryos and larvae were kept at 28 °C in a dark incubator. To prevent pigmentation, 1-Phenyl-2-thiourea (PTU, 0.003% final concentration) was added at 24 h post fertilisation (hpf). Zebrafish were used under the approval of the Animal Welfare Body (OPBA, Organismo Per il Benessere Animale) of the University of Trento and the Italian Ministero della Salute (Project Number 151/2019-PR).

### 2.2. Whole-Mount In Situ Hybridisation

RNA from zebrafish embryos and larvae at different stages was extracted using standard methods (TRIzol, Life Technology Corporation, Carlsbad, CA, USA) and reverse-transcribed using a reverse transcriptase (Super Script II, Life Technology Corporation, Carlsbad, CA, USA). The cDNA was used as a template for a PCR (initiation: 95 °C 5 min; 34 cycles of 95 °C 30 s, 60 °C 1 min, 72 °C 30 s, and finally 72 °C 5 min), using the following primers: *5*′-GACGAGGGCGTCTACACCTG-*3*′ and *5*′-ACACACCCTCGCTTTCCCAA-*3*′ for *negr1* and *5*′-CATCCTCTTCACGGGCAATG-*3*′ and *5*′-CTGAGGAGCGACAGGGTTAA-*3*′ for *opcml*. The PCR products (1027 bp for *negr1* and 811 bp for *opcml*) were purified with Qiagen PCR purification kit and cloned into the pCRII-TOPO vector with the TOPO cloning kit (Invitrogen). Plasmids were linearised and antisense and sense probes for in situ hybridisation were transcribed using the T3/T7 Polymerase (Thermo Scientific, Life Technology Corporation, Carlsbad, CA, USA) and digoxigenin/fluorescein RNA labelling kits (Roche, Basel, Switzerland).

In situ hybridisation was performed according to standard procedures [44]. In short, zebrafish embryos and larvae were fixed with 4% PFA at different stages of development and stored in 100% Methanol at −20 °C for at least 24 h. Rehydration and permeabilisation in ProteinaseK were followed by refixation in 4% PFA and digoxigenin-labelled RNA probe incubation in a water bath at 65 °C overnight. After thorough washing, incubation with anti-dig FAB fragments at 4 °C overnight was performed. On the last day, washing was followed by a colorimetric reaction using BM Purple AP Substrate (Roche) according to standard procedures [44]. Stained larvae were embedded in glycerol and imaged using a Zeiss Axio Imager M2 microscope in brightfield mode using a 10 and 20× objective. For the sections, larvae were embedded in 5% agarose and sectioned using a Leica VT 1200 Vibratome.

### 2.3. PCR

PCR on cDNA of 16-cell stage embryos was performed (initiation: 95 °C 5 min; 34 cycles of 95 °C 30 s, 60 °C 1 min, 72 °C 30 s, and finally 72 °C 5 min) using the following primers: *5*′-ATGGTGTGCAAGCCACTGGA-*3*′ and *5*′-ACGGTTCAACCATGCTCCTT-*3*′ for *negr1* and *5*′-CATCCTCTTCACGGGCAATG-*3*′ and *5*′-CTGAGGAGCGACAGGGTTAA-*3*′ for *opcml*. The expected bands of 399 bp and 811 bp, respectively were purified with a QIAex Purification Kit (Qiagen, Hong Kong, China) and sequenced.

## 3. Results

### 3.1. negr1 and opcml Transcripts Are Maternally Deposited in the Early Embryo

To assess the temporal and spatial expression of *negr1* and *opcml* during embryonic development, we performed whole-mount in situ hybridisation between the 16-cell stage and 5 days post fertilisation (dpf). In the fertilised egg, many maternal gene transcripts are present to facilitate the earliest events in development. Only after a process called midblastula transition (MBT), which in zebrafish occurs at the 512-cell stage (ca. 2 h post fertilisation (hpf)), do the embryos activate transcription [45].

Unexpectedly, given the genes’ reported functions at late stages of development and adulthood in mammals, we revealed ubiquitously distributed maternal mRNA transcripts of both genes already at the 16-cell stage (Figure 1A,B). To confirm this discovery, we additionally performed PCR on retrotranscribed RNA extracted from embryos at the 16-cell stage. Sequence analysis of the resulting RT-PCR products validated the presence of the transcript at this stage of development for both *negr1* and *opcml* (Figure 1E). At 12 h post fertilisation (hpf), strong and specific expression was detected for *negr1* in the midbrain and the otic placodes (Figure 1C). *opcml* showed weak ubiquitous staining with increased intensity in the caudal part of the embryo (Figure 1D).

### 3.2. Similarities and Differences in negr1 and opcml Expression at 24 hpf and 48 hpf

At 24 hpf, both *negr1* and *opcml* exhibited rather weak ubiquitous expression in the central nervous system with several exceptions of distinct expression (Figure 2A–D). Both gene transcripts are present in the olfactory placodes and the pineal gland. Here, the expression appears complementary. *negr1* is expressed in the centrally located cells of the pineal gland, while *opcml* is present in the outer pineal cells (Figure 2A,B insets). Additionally, the otic vesicles exhibited expression of both *negr1* and *opcml* (Figure 2C,D insets). Unlike *opcml*, *negr1* is also expressed in the most caudal part of the spinal cord (Figure 2C upper inset). By 48 hpf, both *negr1* and *opcml* remain expressed in the olfactory bulb and in the hindbrain (Figure 2E–H). In the pineal, *opcml* continues to be expressed, although the expression appears to be weaker in comparison to earlier stages (Figure 2E–H). Conversely, *negr1* starts to be expressed in the ventral telencephalon and the pre-thalamus at this stage (Figure 2E,G and inset in Figure 2E).

### 3.3. negr1 and opcml Expression in the Brain at 96 hpf and 120 hpf

*negr1* and *opcml* continued to be expressed in the olfactory bulb at 4 and 5 dpf (Figure 3). In addition, both genes started to be expressed in the pallium area of the telencephalon. Moreover, we observed transient *negr1* and *opcml* expression in a few cells of the cerebellum, which likely correspond to a subpopulation of Purkinje cells at 4 dpf (Figure 3A,B inset). At 5 dpf, both genes were expressed bilaterally and symmetrically in habenular neurons (Figure 3E,F), while only *negr1* started to be expressed again in cells of the pineal gland (Figure 3E–H). Additionally, *negr1* and *opcml* were both expressed in the inner nuclear layer of the retina (Figure 3G,H).

## 4. Discussion

Our present analysis of the two IgLON family members *negr1* and *opcml* reveals new insights into their spatio-temporal expression dynamics in the developing vertebrate embryo. Initially, both genes are provided maternally in zebrafish. The importance of maternal gene function in neural network formation is well established and evolutionarily conserved in vertebrates and invertebrates [46,47,48]. In mice and rats, NEGR1 and OPCML are needed mainly for neuronal outgrowth, dendritic arborisation, and synapse formation, and it may be interesting to investigate their potential involvement in earlier developmental processes. Furthermore, a maternal function of these genes might account for differences in the severity or onset of NEGR1- or OPCML-linked psychiatric disorders. At subsequent stages of development, both genes are expressed in discrete, partially overlapping domains of the zebrafish brain. Available gene expression data in mammals focused on late developmental stages and adult tissues [36,37,49]. Nevertheless, our data suggest that various expression domains appear conserved between zebrafish and mammals. Notably, some of these domains have been linked to neuropsychiatric disorders which both genes have been implicated in, such as major depressive disorder (MDD), schizophrenia [27,28], and autism spectrum disorder (ASD) [27,28,33]. For instance, *negr1* and *opcml* are strongly expressed in the zebrafish pallium, which has been proposed to harbour structures homologous to the hippocampus and amygdala of mammals [50] involved in ASD and schizophrenia [51]. NEGR1^−/−^ mice exhibit a reduced volume of brain regions, including the hippocampus. Specifically, the parvalbumin-positive interneurons were significantly reduced [34,38]. In OPCML-deficient mice, the hippocampal area develops largely normally, but alterations in hippocampus-dependent spatial learning and memory were observed [52]. In primary hippocampal neurons, the absence of OPCML caused increased numbers of filopodia-like spines and fewer mature spines and neurons, which might explain the behavioural phenotypes [52].

Moreover, we found *negr1* and *opcml* expression in restricted subpopulations of cells in the zebrafish cerebellum, which by position correspond to the Purkinje cells in line with the expression profile found in rodents [53]. The function of these GABAergic projection neurons has also been linked to ASD [54,55]. *negr1* and *opcml* expression in zebrafish is also evident in the dorsal diencephalon. Here, the genes are found both in the medially located pineal gland and in the left and right adjacent habenulae. Interestingly, the pineal gland exhibits complementary expression of the two genes: *negr1*-expressing cells are centrally located while *opcml*-positive cells are located mainly in the outer layer of the pineal gland. These two layers have distinct functions, as the outer layer cells are typically active in periods of darkness [56]. In addition, the regulation of *negr1* expression seems to fluctuate over time: the expression is stronger early in development, decreases subsequently, and increases again at 5 dpf. In contrast, *opcml* expression in the pineal gland is strong at 24 hpf and decreases in the course of development. We did not observe any transcripts in zebrafish parapineal cells, which are known to influence the neurogenesis of the left habenula, causing the left–right asymmetric formation of neuronal subpopulations [57,58]. *negr1* and *opcml* expression in the zebrafish habenulae is symmetric and begins at developmental stages after habenular neuron differentiation [59,60,61]. The habenular neurotransmitter system has been connected to autism and schizophrenia [42,62]. It has also become a major focus for the treatment of MDD. Indeed, patients not responding to conventional pharmacological treatments often benefit from deep brain stimulation to transiently inactivate the lateral habenulae [63,64]. *negr1* and *opcml* expression in the mammalian pineal gland or the habenulae has not been reported to date. It may be revealing to re-analyse the genes in greater detail and to also include earlier stages of brain development. The habenulae relay sensory information such as visual and olfactory input to mid- and hindbrain areas [65,66,67,68]. We found that *negr1* and *opcml* are strongly expressed in the zebrafish olfactory bulb throughout development. This resembles gene expression in the mammalian olfactory system [36,37,49]. Olfaction has recently become a focus in the field of ASD research as it is involved in social behaviour [69]. However, a link between smell and neuropsychiatric disorders in mice mutant for NEGR1 or OPCML remains to be explored.

Our detailed expression study extends and refines currently available expression data on zebrafish *negr1* and *opcml* [44]. It should encourage future research to, for instance, analyse mammalian NEGR1 and OPCML expression and function in brain areas and at developmental time points other than those described to date. A particular area of interest may be to investigate a maternal contribution of the two genes and their expression and function in the habenular neural circuit. These studies could reveal new insights into the onset of neuropsychiatric disorders, their severity and the brain area(s) affected. A zebrafish knock-out for *negr1,* which did not result in overt morphological alterations, has been reported in the framework of a large schizophrenia study [70]. In the same study, an *opcml* mutant was generated, which, however, was not further described. For both gene knock-outs, no functional or behavioural investigations are available. Our detailed expression analysis should provide an excellent starting point for such in vivo studies, for instance to determine the precise temporal role of the genes during neural network formation and function. This in turn provides a platform for expression profiling to unravel the molecular network downstream NEGR1 and OPCML for the identification of suitable therapeutic targets. Furthermore, it aids screening for disorder-ameliorating compounds, which subsequently can be tested in mammalian models. Combining the advantages of available resources to gain deeper insights into the mechanisms underlying NEGR1 and OPCML function in health and disease is a pivotal prerequisite for developing therapies.

## Figures and Tables

**Figure 1 genes-15-00363-f001:**
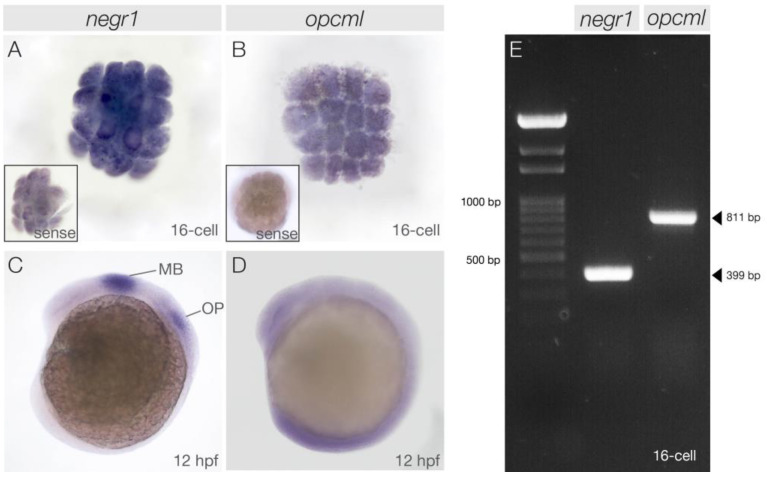
*negr1* and *opcml* expression at early developmental stages. 16-cell stage zebrafish embryos expressing *negr1* (**A**) and *opcml* (**B**); sense probe stains are shown in the insets. (**C**,**D**) show lateral views of 12 hpf old embryos. *negr1* is expressed in the midbrain and the otic placodes (**C**), while *opcml* exhibits ubiquitous expression (**D**). RT-PCR corroborates the presence of *negr1* and *opcml* transcripts at the 16-cell stage (**E**). MB, midbrain; OP, otic placodes.

**Figure 2 genes-15-00363-f002:**
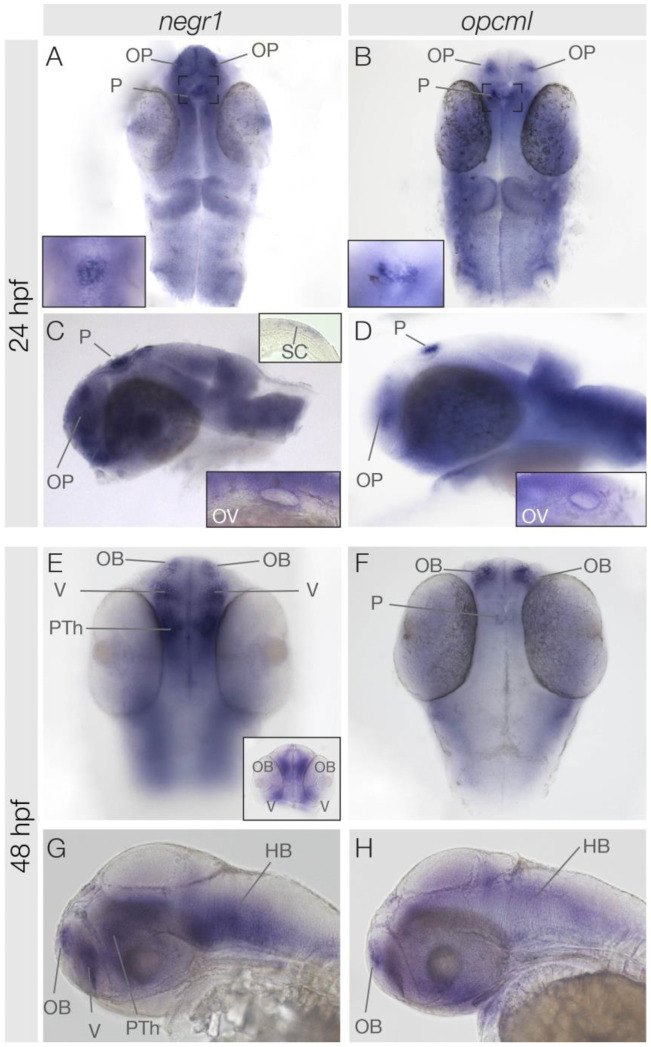
Expression patterns of *negr1* and *opcml* at 24 hpf and 48 hpf. Dorsal (**A**,**B**,**E**,**F**) and lateral (**C**,**D**,**G**,**H**) views of the head region of zebrafish embryos at stages indicated. Insets show the pineal gland (**A**,**B**), the otic vesicle (**C**,**D**), the spinal cord (upper inset in **C**), and a frontal view (**E**). HB, hindbrain; OB, olfactory bulb; OP, olfactory placode; OV, otic vesicle; PTh, pre-thalamus; P, pineal gland; SC, spinal cord; V, ventral telencephalon.

**Figure 3 genes-15-00363-f003:**
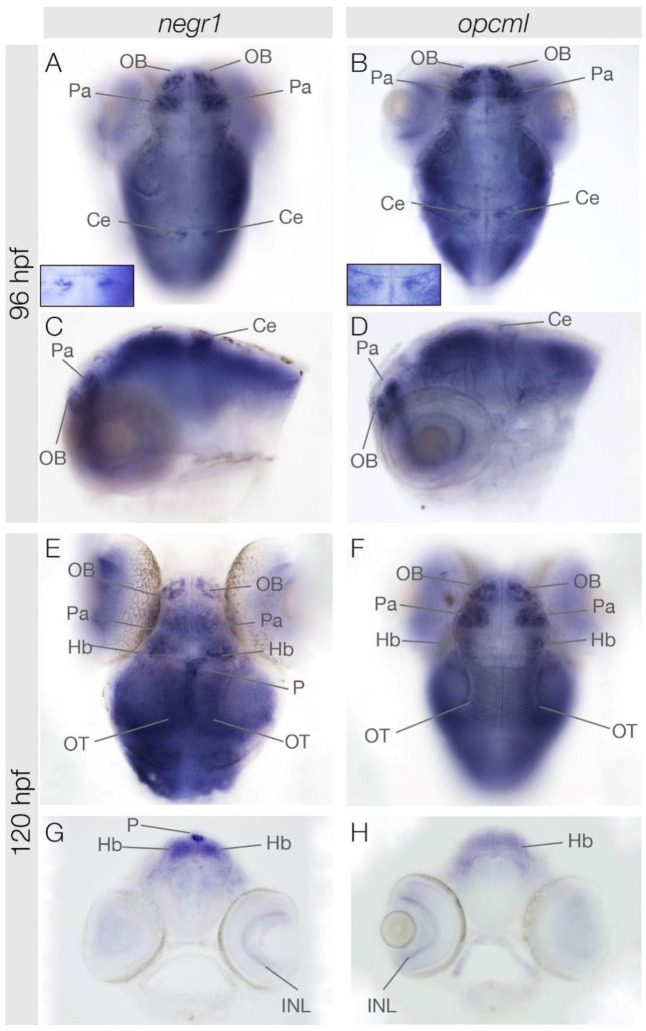
Expression patterns of *negr1* and *opcml* at 96 hpf and 120 hpf. Dorsal (**A**,**B**,**E**,**F**) and lateral (**C**,**D**) views of zebrafish larvae focussed on the head at stages indicated. Transversal sections were carried out at the level of the pineal organ (**G**,**H**). Ce, cerebellum (inset in (**A**,**B**)); Hb, habenula; INL, inner nuclear layer; OB, olfactory bulb; Pa, pallium; P, pineal gland; OT, optic tectum.

**Table 1 genes-15-00363-t001:** Gene and protein similarity for *negr1* and *opcml* in zebrafish and mammals, including previous gene names.

	**Human (*Homo sapiens*)** NEGR1 (IgLON4, KILON, NTRA)	**Mice (*Mus musculus*)** Negr1 (Ntra, neurotractin)	**Rat (*Rattus norvegicus*)** Negr1
**Zebrafish (*Danio rerio*)** *negr1*	64.1% similarity (bp sequence) 78.1% similarity (aa sequence)	61.7% similarity (bp sequence) 75.8% similarity (aa sequence)	63% similarity (bp sequence) 75.8% similarity (aa sequence)
	**Human (*Homo sapiens*)** OPCML (IGLON1, OBCAM, OPCM)	**Mice (*Mus musculus*)** Opcml (Obcam)	**Rat (*Rattus norvegicus*)** Opcml
**Zebrafish (*Danio rerio*)** *opcml*	58.7% similarity (bp sequence) 78.3% similarity (aa sequence)	61.4% similarity (bp sequence) 77.8% similarity (aa sequence)	62.2% similarity (bp sequence) 77.8% similarity (aa sequence)

## Data Availability

Data supporting reported results can be requested.

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
