# Peer review of "The Risk Genes for Neuropsychiatric Disorders negr1 and opcml Are Expressed throughout Zebrafish Brain Development"

_genes, 2024, doi:10.3390/genes15030363_

Round 1

Reviewer 1 Report

Comments and Suggestions for Authors

This study focuses on the spatio-temporal expression dynamics of NEGR1 and OPCML during early zebrafish embryonic development. The authors characterized the expression patterns of these IgLON family members from the 16-cell stage to 5 days post-fertilization using whole-mount in situ hybridization and PCR analyses. Their findings revealed novel insights, including the maternal provision of NEGR1 and OPCML in zebrafish embryos and their expression in partially overlapping domains throughout brain development. Notably, the expression patterns observed in specific brain regions parallel those implicated in neuropsychiatric disorders in mammals, suggesting potential relevance to the pathophysiology of these conditions.

However, the rationale behind the study and the real importance of these results remain somewhat unclear. Further clarification on the specific hypotheses driving this investigation and the potential implications of the findings for our understanding of neuropsychiatric disorders would greatly enhance the significance of the work.

Questions for the authors:

This study aims to address the molecular underpinnings of diseases and these genes in development. To what extent is the use of zebrafish the best approach in understanding human disease? Could the authors explain the rationale behind the choice of the study model?

The use of PTU for enhanced clarity of zebrafish is a common practice, however, recent reports show that this compound can alter autophagy during neural development (PMID: 32286915v). Considering that this is a pivotal cellular process governing several molecular events, namely gene expression, how did the authors account for this potential interference?

How do the observed expression patterns of NEGR1 and OPCML in zebrafish embryos compare with existing data on mammalian expression, particularly in brain regions associated with neuropsychiatric disorders?

How do the authors envision the findings of this study contributing to our understanding of neuropsychiatric disorders associated with NEGR1 and OPCML mutations? Are there specific implications for potential therapeutic interventions?

Could the authors elaborate on the potential functional implications of maternal provision of NEGR1 and OPCML in zebrafish embryos? How do you think that this might be correlated with a predisposition for neuropsychiatric disorders?

Considering the expression of NEGR1 and OPCML in distinct brain regions, what are your hypotheses regarding their specific functional roles in these regions, and how might these roles contribute to the onset or progression of neuropsychiatric disorders?

Considering the conserved nature of IgLON protein-encoding genes across species, do the authors plan to investigate the functional implications of NEGR1 and OPCML expression patterns observed in zebrafish embryos in mammalian models?

In light of the proposed future research directions, what experimental approaches do you believe would be most promising for further elucidating the functional roles of NEGR1 and OPCML in health and disease, particularly in the context of neuropsychiatric disorders?

Comments on the Quality of English Language

The language appears to be generally appropriate for an academic paper. There are a few instances where the language could be further refined for clarity and precision. Additionally, there are a few grammatical errors and awkward constructions that could benefit from proofreading and editing for smoother presentation. Overall, while the language is largely appropriate, some areas would benefit from minor proofreading and refinement to enhance clarity and coherence.

Reviewer 2 Report

Comments and Suggestions for Authors

In this study Authors aimed to analyze the expression of two genes, known to be implicated in neuropsychiatric conditions, in developing zebrafish.

This is very elegant study, photos are well done, but I would include bigger photos to make them more visible. Also, please explain IgLON in abstract- I did not know this, so I assume some other Readers may also.

Line 52: its unclear what Authors meant here, please re-write

Please provide also strain for zebrafish line because its missing.

Please, add small table with comparison of zebrafish genes with human, mica and rat genes (to show similarity). 
